# Spectral phase control of interfering chirped pulses for high-energy narrowband terahertz generation

Spencer W. Jolly [1,2,7], Nicholas H. Matlis[3,7], Frederike Ahr[3], Vincent Leroux [1,2], Timo Eichner[1], Anne-Laure Calendron [3,4], Hideki Ishizuki[5,6], Takunori Taira[5,6], Franz X. Kärtner [3,4] & Andreas R. Maier [1]

Highly-efficient optical generation of narrowband terahertz radiation enables unexplored technologies and sciences from compact electron acceleration to charge manipulation in solids. State-of-the-art conversion efficiencies are currently achieved using difference-frequency generation driven by temporal beating of chirped pulses but remain, however, far lower than desired or predicted. Here we show that high-order spectral phase fundamentally limits the efficiency of narrowband difference-frequency generation using chirped-pulse beating and resolve this limitation by introducing a novel technique based on tuning the relative spectral phase of the pulses. For optical terahertz generation, we demonstrate a 13-fold enhancement in conversion efficiency for 1%-bandwidth, 0.361 THz pulses, yielding a record energy of 0.6 mJ and exceeding previous optically-generated energies by over an order of magnitude. Our results prove the feasibility of millijoule-scale applications like terahertz-based electron accelerators and light sources and solve the long-standing problem of temporal irregularities in the pulse trains generated by interfering chirped pulses.

[1] Center for Free-Electron Laser Science and Department of Physics Universität Hamburg, Luruper Chaussee 149, 22761 Hamburg, Germany. [2] Institute of Physics of the ASCR, ELI-Beamlines project, Na Slovance 2, 18221 Prague, Czech Republic. [3] Center for Free-Electron Laser Science and Deutsches Elektronen Synchrotron (DESY), Notkestraße 85, 22607 Hamburg, Germany. [4] Department of Physics and The Hamburg Centre for Ultrafast Imaging, Universität Hamburg, Luruper Chaussee 149, 22761 Hamburg, Germany. [5] Division of Research Innovation and Collaboration, Institute for Molecular Science, 38 Nishigonaka, Myodaiji, Okazaki, Aichi 444-8585, Japan. [6] Innovative Light Sources Division, RIKEN SPring-8 Center, 1-1-1 Kouto, Sayo-cha, Sayo-gun, Hyogo 679-5148, Japan. [7] These authors contributed equally: Spencer W. Jolly, Nicholas H. Matlis. Correspondence and requests for materials should be addressed to S.W.J. (email: spencer.jolly@cea.fr)

Recent years have seen a tremendous surge in development of laser-based, high-field terahertz (THz) sources for a wide range of applications, including THz-based accelerators[1–3], control and metrology of electrons[4], femtosecond X-ray pulse characterisation[5,6], control of dynamics in solids[7,8] and spectroscopy[9]. Most work has concentrated on generation of single-cycle, broadband THz pulses, resulting in a mature technology yielding close to millijoule energies and optical-to-THz conversion efficiencies near 1%[10–13]. Some emerging applications, however, such as waveguide-based electron acceleration[14], resonant pumping of materials[15] and narrowband spectroscopy[9], require high-field, narrow-bandwidth THz pulses for which the technology is much less developed. In particular, relativistic acceleration of electrons calls for optically synchronised multi-millijoule pulses with MV/cm electric-field strengths, frequencies below 1 THz and bandwidths below 1%[16,17], for which no technology currently exists.

Early on, periodically-poled lithium niobate (PPLN) was identified as promising for optical difference-frequency generation (DFG) of multicycle THz pulses[18,19] due to its high nonlinear coefficient, $d_{33}$ ~25 pm/V, and predicted conversion efficiencies in the range of several percent[17]. Such crystals would enable generation of THz pulses with tens-of-millijoule energies, sub-percent bandwidths and MV/cm fields using the Joule-level optical pulses available today. However, the insufficient optimisation of the narrowband optical-to-THz conversion, and the resultant low fraction of optical photons that contribute to this process, is a primary obstacle to achieving THz pulses with the above parameters. Recent work has therefore concentrated on addressing the systematic optimisation of laser and crystal parameters with a view towards making every photon count.

Initial experiments using compressed Ti:Sapphire pulses demonstrated that the THz yield can be significantly increased (×5) by optimising the optical bandwidth of the driver laser and by cryogenic cooling of the PPLN which reduces THz reabsorption[20]. Using short-pulse drivers, however, results in high intensities that limit the energy that can be applied to the crystal before the conversion efficiency saturates or the crystal damages. In addition, the simultaneous presence of photons spanning a broad bandwidth enables phase-matched generation of unwanted, higher-frequency THz[20], which not only reduces efficiencies by robbing energy from the main process, but also contributes to damage and thus reduces the total amount of useful pump energy that can be applied to the crystal.

These issues, however, can be overcome by pumping a nonlinear crystal with temporally chirped pulses. Chirping the drive pulse reduces the peak intensity, which allows for large increases in optical energy, and also reduces the instantaneous spectral bandwidth by mapping frequency onto time. In order to provide the correct spectral content for DFG, two chirped pulses must then be combined with a delay.

This chirp-and-delay technique, pioneered originally for driving photoconducting antennae[21], has since been extensively used in a broad range of scientific settings[22–24] because it provides a simple means of creating optical pulse trains of tunable number and periodicity which can be effectively used to drive resonant processes. A complication is that chirped pulses from chirped-pulse amplification (CPA) based laser systems inherently contain high-order phase, such as third-order dispersion (TOD), which induces temporal variations in the periodicity of the pulse train (temporal beating) and can reduce effectiveness of the driver[23,25]. For narrowband THz generation, the TOD-induced pulse-train periodicity variations, which are connected to temporal variations in the instantaneous difference-frequency content, can severely limit the conversion efficiency. Unfortunately, the intuition to completely remove the TOD from the chirped drive pulses leads to approaches which are either practically infeasible or produce undesired changes in the pulse properties.

Here, we present an elegant and easy-to-implement method for precisely controlling the temporal variation of the optical modulation period. Instead of removing the TOD, we simply tune the relative spectral phase between the chirped pulses. The former linearises the chirp, but requires an impractically large change to both pulses, while the latter uses a very small difference in chirp between the pulses to monochromatise the difference-frequency spectrum and regularise the pulse train. Our method not only enables significant optimisation for efficiently driving narrowband resonant processes but also offers new possibilities in producing designer pulse trains with varying periodicities optimised for specific purposes. We apply this technique to the problem of optimising the conversion efficiency for narrowband, multicycle THz generation by regularising the driver pulse train. By tuning the relative spectral phase in a chirp-and-delay setup, we achieve a 13-fold relative enhancement in conversion efficiency resulting in sub-percent bandwidth, 361 GHz pulses with a conversion efficiency of 0.24%. Using a large-aperture PPLN, we demonstrate a record energy of 0.6 mJ which exceeds previously reported energies by over an order of magnitude[26,27].

## Results

**Monochromatising the difference-frequency spectrum**. The idealised case for narrowband difference-frequency generation (DFG) calls for two narrow-line optical sources with a well-defined frequency difference, $\Delta\omega$. Unfortunately, these sources are not yet available with appreciable energies. Chirp-and-delay offers a way of obtaining, effectively, a two-line narrowband source from a broad-band, linearly chirped laser by combining two pulses with a delay $\Delta t$. The resulting instantaneous spectral features are narrow and have a constant frequency difference $\Delta\omega$. For this reason, chirp-and-delay is also known as spectral focusing[28]. By adjusting the delay, $\Delta\omega$ can be tuned to match the design frequency $\Omega_{THz}$ of a PPLN crystal, enabling effective conversion of the provided difference-frequency content into a THz pulse (Fig. 1a).

A complication of this concept is that chirped pulses from CPA-based laser systems include not only group delay dispersion (GDD), i.e., linear chirp, but also non-negligible third-order dispersion (TOD) which adds a quadratic component, i.e., curvature, to the chirp (Fig. 1b). The result is a linear variation of the difference frequency with time (i.e., difference-frequency chirp) which detunes a fraction of the drive pulse energy away from the resonance condition (e.g., phase-matching) of the driven process. This effect becomes increasingly severe as the bandwidth of the resonance narrows, such as for difference frequency generation of narrowband THz pulses with a large number of cycles.

An intuitive solution would be to completely remove the TOD, thereby linearising the chirp. Unfortunately, this solution is generally impractical because most dispersive systems couple TOD with large amounts of GDD and thereby also produce undesired changes to the pulse duration. Although chirped mirrors can be designed to provide pure TOD, they typically compensate only of order $10^4$ fs$^3$ per bounce, while CPA systems usually have at least $10^6$ fs$^3$ of TOD. It is, however, not necessary to completely remove the TOD. What is actually important is to monochromatise the difference frequency, $\Delta\omega$, which can be achieved much more elegantly by simply adding a small amount of spectral phase to one of the two pulses (see Methods). The effect is to slightly tilt one pulse in time-frequency space (Fig. 1c) so that the temporal dependence of $\Delta\omega$ is eliminated despite the chirp curvature. In this way, $\Delta\omega$ remains within the narrow

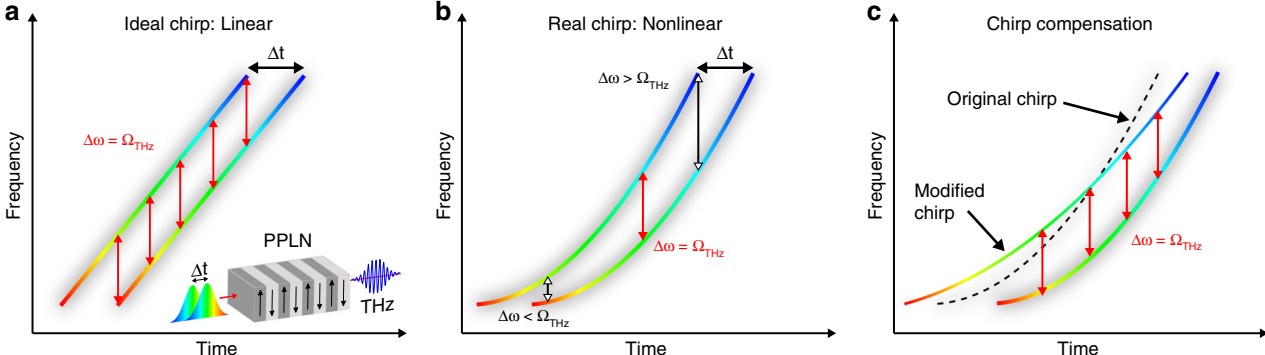

**Fig. 1** Chirp-and-delay concept. **a** Two broadband but linearly chirped and delayed pulses provide narrow spectral features with constant instantaneous difference frequency $\Delta\omega$, which is converted in a PPLN crystal into a THz pulse of frequency $\Omega_{THz} = \Delta\omega$. **b** Due to higher order dispersion $\Delta\omega$ varies along the pulse and limits the range where the provided difference frequency fulfils the phase matching condition in the PPLN. **c** Slightly tilting one of the pulses in time-frequency space regularises $\Delta\omega$ and thereby maximises THz generation

acceptance bandwidth of the PPLN around $\Omega_{THz}$ over the full duration of the pulse, optimising the efficiency of the process.

Equivalently, this optimisation of the THz conversion efficiency can be described in the time domain: when the periodicity of the optical intensity modulation matches the THz carrier period, the THz generated by successive pulses in the train add coherently. Irregularities in the pulse-train periodicity induce destructive interferences which become more severe as the number of THz cycles increases. Monochromatising the difference frequency $\Delta\omega$, which determines the temporal beat frequency, ensures a regular pulse train and coherent addition of THz waves.

**Millijoule-scale narrowband THz generation.** To demonstrate this concept experimentally, we use a modified Mach-Zehnder interferometer (shown in Fig. 2a) in which uncompressed pulses from a Ti:Sapphire system are split into two identical copies and recombined after spectral phase manipulation.

Typical CPA systems have a positive chirp curvature, so monochromatising $\Delta\omega$ requires adding positive GDD and negative TOD (see Methods). However, negative TOD is generally not available. While it is conceptually simpler to apply dispersion compensation to only one pulse, as illustrated in Fig. 1, an equivalent solution is to add GDD to one pulse and TOD with reversed sign to the other. We therefore split the dispersion compensation between the two interferometer arms, which allows us to use custom-designed dispersion compensating mirrors (DCMs), available with positive TOD, to tune the second pulse. This design also compactifies the setup. A tunable amount of GDD is added by inserting a variable thickness of high-dispersion glass (SF11) and TOD is added incrementally by replacing high-reflectors with DCMs.

In order to maximise the pump energy that we can apply to the DFG process, we employ large-aperture (shown in Fig. 2b), magnesium-doped PPLN (LA-PPMgLN) crystals which were custom fabricated by the Institute for Molecular Science in Japan[29]. Since Mach-Zehnder interferometers necessarily provide two identical outputs, we send each onto independent, cryogenically cooled crystals. This configuration provides a natural way to effectively double the crystal aperture, further increasing the allowable incident energy and making use of energy that would otherwise be wasted. Total optical pulse energies in excess of 1 Joule are thus safely applied to the crystals. The resulting THz pulse energy is then measured with a pyroelectric detector and the spectrum is characterised using a longitudinal interferometer.

Here we investigate the performance of crystals with a poling period of 330 µm and a corresponding emission frequency of 361 GHz, which is ideally suited for powering a THz-based accelerator[16]. Details of the performance of a PPLN crystal designed for a higher frequency of 558 GHz (i.e., with a periodicity of 212 µm), are presented in Supplementary Note 1. For our driver laser, which has a GDD of $2.05 \times 10^6$ fs² and a TOD of $-4 \times 10^6$ fs³, we calculate (see Methods) an optimum compensation of $\Delta GDD = 9100$ fs² and $\Delta TOD = 54,000$ fs³ for the 330 µm crystal. Note that these phase compensation amounts are much smaller than the original spectral phase (i.e., $\Delta GDD \ll GDD$ and $\Delta TOD \ll TOD$) which ensures the pulse duration is not significantly affected and is a key factor in the practicality of our technique.

Figure 3a shows the THz signal as a function of the delay between the driver pulses. In the uncompensated case (red dots), THz is generated with low signal levels over a broad range of delays. Here, each delay providing THz signal corresponds to matching of the instantaneous frequency difference, $\Delta\omega$, with the PPLN design frequency, $\Omega_{THz}$, at a temporally and spectrally distinct part of the pulse. Adding GDD by increasing the amount of SF11 in the beam enhances the THz signal (blue triangles) by allowing more parts of the pulse to contribute to THz generation simultaneously. As a result, the range of THz-generating delays narrows until most of the pulse is engaged at a single delay. The experimentally determined optimum (green squares) of $\Delta GDD = 9200$ fs² (or 49 mm SF11) is in very good agreement with our calculation and results in an 11-fold increase in THz output.

Using DCMs to tune the relative TOD between the two driver pulses, the THz signal is further enhanced by 20% yielding in total a 13-fold increase in output compared to the initial configuration (Fig. 3b). The measured optimum of $\Delta TOD = 20,000$ fs³ is different from the predicted optimum, which may be attributable to the presence of fourth and higher orders in the pump spectral phase. In contrast to the GDD correction, the discrete nature of the TOD compensation makes fine-tuning of the TOD difficult.

The above optimisation of the THz yield can be understood as an optimisation of the temporal range over which the time-dependent frequency difference $\Delta\omega(t)$ falls within the THz bandwidth $\Omega_{THz} \pm \Delta\Omega_{THz}$ accepted by the phase-matching process (Fig. 4a). The portion of the optical pulse contributing to THz generation is then simply estimated as the fraction satisfying $|\Delta\omega(t) - \Omega_{THz}| \le \Delta\Omega_{THz}$. More precisely, this fraction is the integrated product of the phase-matching efficiency curve in the time domain and the temporal intensity profile of the optical pulse (Fig. 4a). This quantity serves as a good measure predicting the variation in THz output with delay and with GDD and TOD corrections, such as the enhancement of THz at

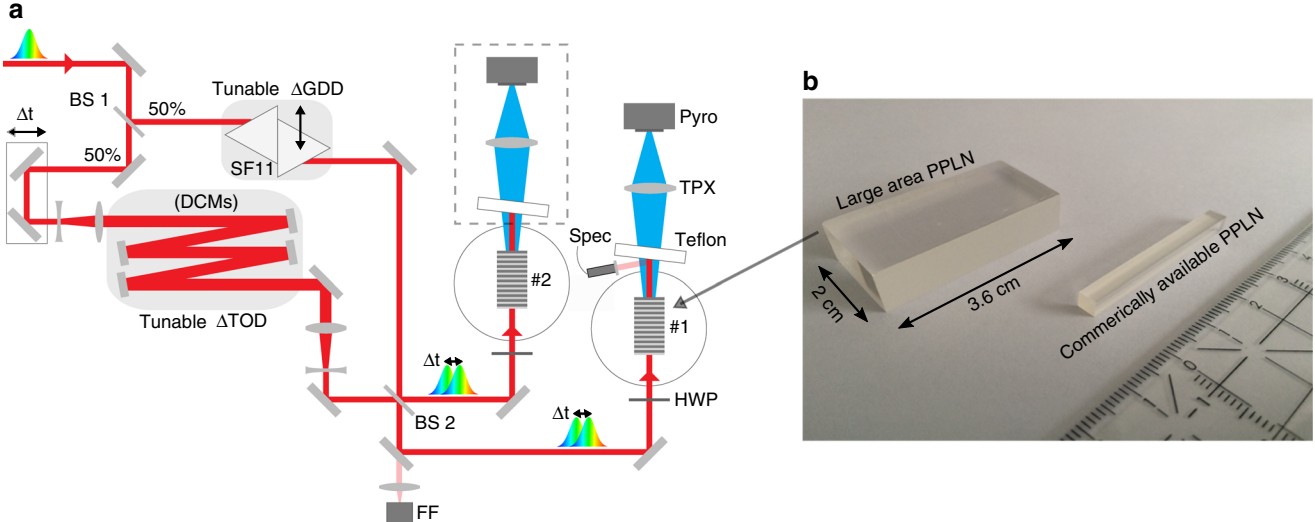

**Fig. 2** Narrowband THz setup. **a** The driver laser output is split into two identical copies, whose GDD and TOD are independently tuned using a variable amount of high-dispersive SF11 glass (adding GDD), and a set of dispersion compensating mirrors (DCM), each adding a discrete amount of TOD. The recombined pulses generate narrowband THz pulses in large-aperture cryogenically cooled magnesium-doped PPLN crystals, shown in the photograph (**b**). To characterise the THz pulses, a pyroelectric detector is used either directly (pulse energy) or as the detector in a longitudinal interferometer (frequency). BS: beam splitter; FF: far field camera; HWP: half-wave plate

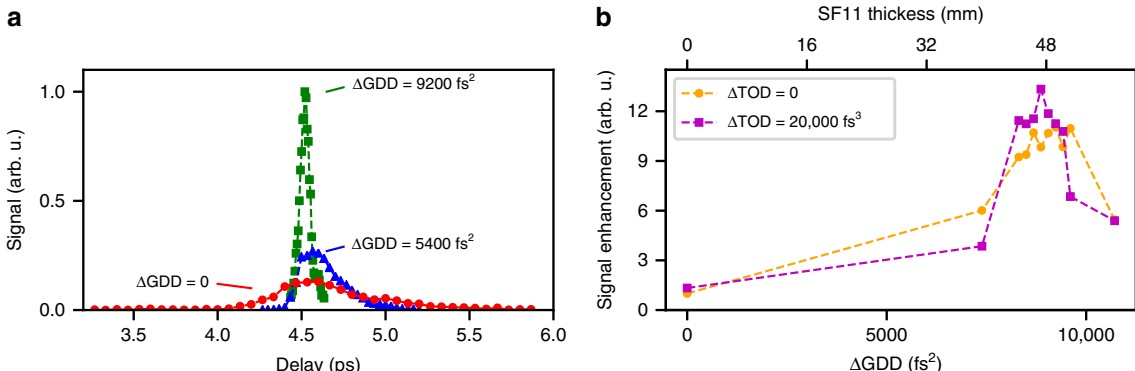

**Fig. 3** Optimisation of the relative spectral phase. **a** Scan of pulse delay to match $\Delta\omega$ to the PPLN frequency $\Omega_{THz}$. Adding $\Delta$GDD, by inserting SF11 into one of the interferometer arms, the THz signal increases up to a factor of 11 (green squares), and the range of THz-generating delays narrows. **b** Conversion efficiency enhancement vs $\Delta$GDD for varying amounts of $\Delta$TOD. For an optimum $\Delta$TOD, the signal increases by another 20% relative to no $\Delta$TOD correction, yielding a total increase in THz output by a factor of 13

positive delays and the reduction of THz at negative delays (Fig. 4b).

To characterise the performance of the crystal, we use two definitions of conversion efficiency which highlight different quantities. The "extracted" efficiency, plotted in Fig. 5a, represents the ratio of useful THz energy extracted from the crystal to the driver incident energy. The peak extracted efficiency of 0.15% is achieved at a fluence of 180 mJ/cm². This figure corresponds to an "internal" efficiency of 0.24%, which represents the ratio of THz and optical energies within the crystal. The internal efficiency serves as a useful metric for the optimisation of the DFG process independent from the practical complexities of input and output coupling. The fine-tuning of the TOD not only increases the conversion efficiency, but also increases the fluence at which optimum conversion efficiency occurs, leading to large increases in the total THz yield.

Figure 5b shows the corresponding extracted THz energies. The highest individual THz pulse energy of 458 μJ and the highest average 2-crystal output of 0.6 mJ are achieved at a fluence of 350 mJ/cm². The uniform red-shift of the optical spectrum (inset)

shows that pump depletion has occurred across the entire spectrum, which confirms that the full optical pulse engages in the DFG process. The spectrum of the generated radiation is measured using an interferometric autocorrelation of the THz pulse (Fig. 5c). The Fourier transform of this trace shows a central frequency of 361 GHz and confirms a sub-percent bandwidth (Fig. 5d). If the pulses were to be combined, the 0.6 mJ and roughly 200 ps pulses could produce a field strength of ~18 MV/m when focused to a $1/e^2$ waist of 2 mm ($2.4 \times \lambda_{THz}$).

Note that the pump fluence for peak yield is well above the fluence for peak conversion efficiency, which suggests benefits from even larger aperture crystals. The difference in performance of the two crystals (Fig. 5b) is partly due to anti-reflection measures, applied to only one crystal, that reduce Fresnel losses and improve the extraction efficiency by 35% (see Methods).

Our measurements show that the need for spectral-phase tuning is driven by the narrow acceptance bandwidth of the process. For a similar degree of tuning, we therefore expect a lower conversion efficiency for a crystal with a narrower phase-matching bandwidth. In fact, this effect is confirmed by the lower

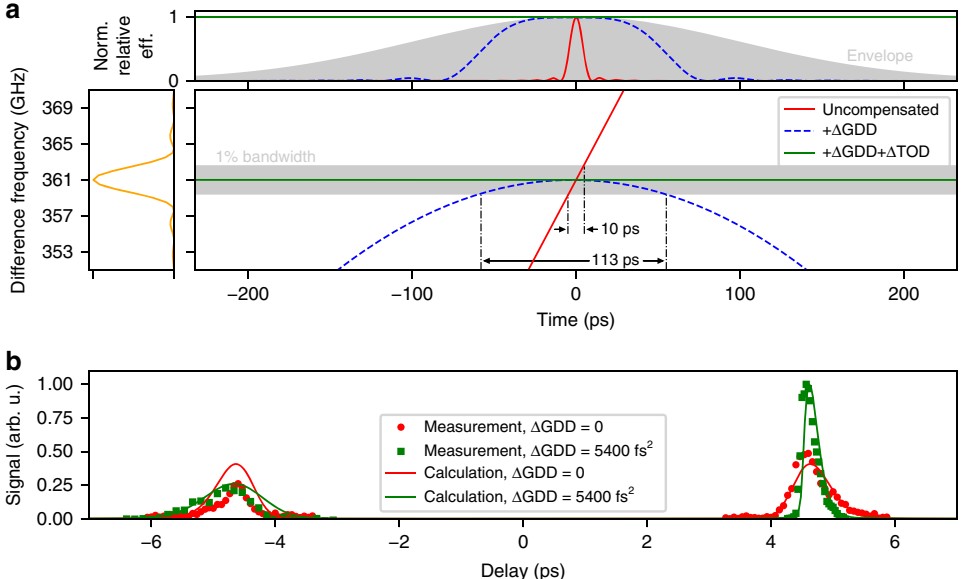

**Fig. 4** Asymmetric chirp compensation. **a** The 330 μm poling period PPLN has a central frequency of $\Omega_{THz} = 361$ GHz with a 1% bandwidth. Initially, the difference frequency $\Delta\omega$ varies linearly in time (red). As spectral phase compensation is applied adding $\Delta$GDD (dashed blue) and further adding $\Delta$TOD (solid green) the fraction of the pulse falling within the THz bandwidth $\Omega_{THz} \pm \Delta\Omega_{THz}$, and thus the generated THz energy, increases. The top inset shows the PPLN bandwidth multiplied by the difference frequency. The integrated product of this phase-matching efficiency with the pulse envelope (grey) is a measure for the fraction of the pulse contributing to THz generation. **b** Adding $\Delta$GDD (green) creates an asymmetry between the two pulses in time that reduces the THz signal at negative delays and enhances the THz at positive delays. For identical pulses (red) the THz signal is equal for positive and negative delays. A calculation (solid lines) based on the concept discussed in the main text and illustrated in panel **a** predicts the THz output in good agreement with the measurements (squares, dots)

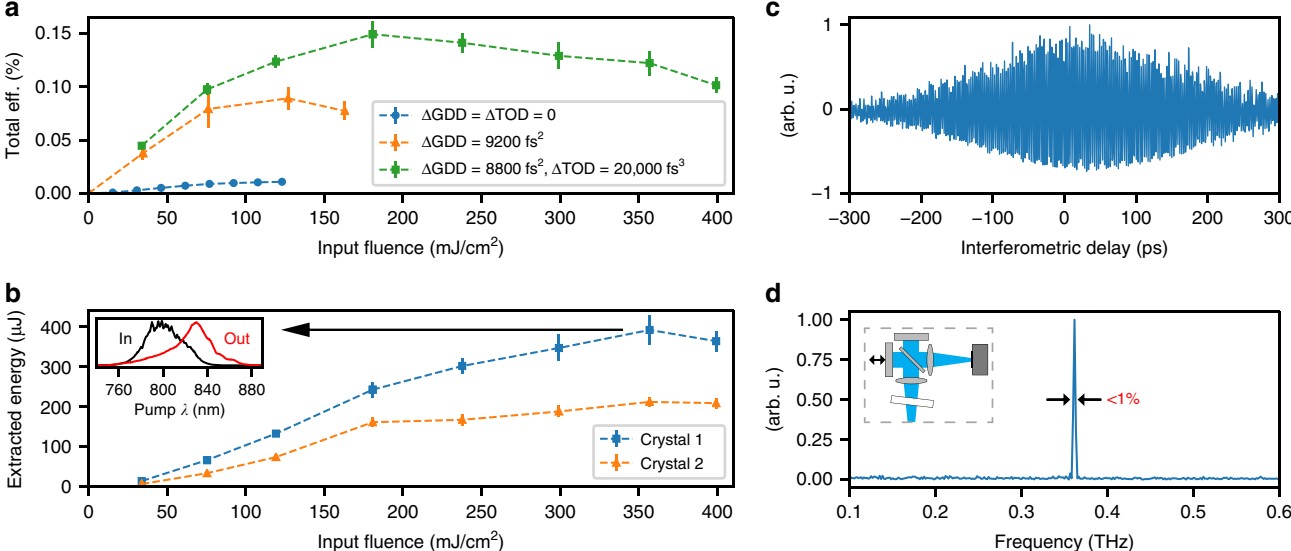

**Fig. 5** High THz energy. **a** Fine tuning the spectral phase increases the fluence at which the maximum conversion efficiency occurs and reaches 0.15% at 180 mJ/cm² for crystal #1. Note, that with simultaneous GDD and TOD correction, the optimum determined $\Delta$GDD is slightly different than the value reported in Fig. 3a. **b** Using GDD and TOD compensation, an input fluence of 350 mJ/cm² generates individual THz pulse energies up to 458 μJ (392 μJ average) and 232 μJ (212 μJ average) in crystal #1 and #2, respectively. The difference in THz output is attributed to differences in the preparation of the crystal surfaces (see Methods). Significant red-shift of the driver spectrum is observed for high-energy THz generation indicating a high degree of optical cascading. **c** Interferometric correlation of the generated THz pulses and its frequency spectrum (**d**), confirming the central frequency of 361 GHz and a sub-1% bandwidth. The interferometer (inset) replaces the energy measurement setup, Fig. 2a. Note in **c** that each data point is averaged over 10 shots. The sampling of the frequency, however, accentuates the visual effect of the shot-to-shot fluctuations, while indeed the centre frequency is captured with high precision (see Supplementary Note 2). Error bars correspond to the standard deviation of 20 measurements in the case of **a** and **b**

efficiency measured with the 212 μm periodicity crystal which has a larger number of poling cycles and therefore a narrower acceptance bandwidth. Accounting for variations in absorption coefficient, crystal length and poling period, we predict a conversion efficiency of 0.08% at 558 GHz which is in very good agreement with the measured value of 0.07% (see Supplementary Note 1).

Despite the large efficiency enhancements obtained by the relative spectral phase tuning, the maximum absolute conversion efficiency reported here is only twice larger than that reported by us previously[27]. An exact comparison between these results is not possible, however, since nonlinear crystals of different aperture, length and vendor were used.

In general, a proper comparison of efficiencies across different experiments requires a careful evaluation of the effects of differences in the laser, THz and crystal parameters, which can be very complex (see Supplementary Note 3). For example, the DFG efficiency intrinsically scales quadratically with THz frequency, due to higher THz photon energies, so one can expect higher efficiencies for higher frequencies. On the other hand, THz absorption also increases with frequency, mitigating, to varying degrees, the photon-energy effect. Differences in the crystal length, poling period, aperture and crystal manufacturer all also strongly affect the conversion process. The effect of crystal length on efficiency is especially complex, since length governs not only the THz bandwidth and therefore the sensitivity of the conversion process to TOD, but also the growth of cascading[17] and amounts of pump depletion and THz reabsorption. Differences in the crystal aperture can lead to significant variations in efficiency via inhomogeneities in the poling structure[29]. The effects of crystal material and poling quality on multicycle THz generation are so far nearly unexplored. A detailed characterisation of these effects is thus an important part of the further optimisation of the multicycle THz generation process.

What we unambiguously demonstrate here is the role of TOD in detuning narrowband THz generation using the chirp-and-delay setup. The degree of relative enhancement achievable by tuning the relative spectral phase is robustly predictable from the amount of TOD and the phase-matching bandwidth, independent of other factors which may affect the efficiency. Mitigation of the effects of TOD is thus a required ingredient for optimising narrowband THz generation using chirp-and-delay.

## Discussion

We demonstrate, for the first time, the sensitivity of narrowband THz generation to TOD in the chirp-and-delay setup and show that this sensitivity can be attributed to temporal variations in the difference-frequency content which cause a very strict phase-matching condition not to be satisfied for the majority of the optical pulse. The effect of TOD can be understood in the time domain as inducing pulse-train irregularities which cause a destructive superposition of THz waves generated by the various pulses within the train. To address this issue, we introduce a powerful new technique based on relative spectral-phase tuning of the chirped pulses. The technique provides, for the first time, a simple and easily-implemented method to control and fine tune the temporal dependence of the difference-frequency spectrum, and hence the pulse-train periodicity, without changes to the laser architecture or to the pulse-train intensity and duration.

For high-energy narrowband THz generation, where the priority is optimising the optical-to-THz conversion efficiency, we adjust the spectral phase to monochromatise the difference-frequency spectrum and to create a regularised pulse-train (see Supplementary Notes 4 and 5), and thus obtain a 13-fold enhancement in THz yield (at 361 GHz). By using a Joule-class

laser as the chirped-pulse source to pump two custom, large-aperture PPLN crystals (LA-PPMgLN[29]), we achieve a record energy of 0.6 mJ which exceeds previous results in optical generation of multicycle THz by over an order of magnitude. The internal conversion efficiency we measure of 0.24% is also nearly twice larger than previously reported for narrowband sources[20,26,27]. This result is especially significant in light of the phase-matching difficulties associated with sub-percent THz bandwidths and the additional complexities associated with scaling to centimetre apertures and Joule-level pump energies[29].

Millijoule-scale narrowband THz pulses represent a breakthrough which demonstrates the feasibility of the emerging technology of THz-based acceleration that holds high promise for revolutionising electron accelerators and the light sources based on them[16]. It is expected that 0.6 mJ of THz in a 200-ps-long pulse as produced in this work could create a field strength of 50 MV/m when coupled in to a waveguide designed for matched acceleration. With acceleration over the roughly 4 cm allowed by such a pulse, sub-relativistic electrons could be accelerated beyond 2 MeV, an energy already relevant for ultrafast electron diffraction. This can be compared to past acceleration in a waveguide with broadband THz, where the acceleration length was limited to 3 mm and the energy gain was limited to 7 keV[1].

However, our results are equally applicable for enhancing the yield of narrowband THz pulses at any energy level. For the average researcher with a mJ- or sub mJ-scale laser system, making every photon count is an imperative. Order-of-magnitude-scale efficiency enhancements are thus a game changer for enabling experiments requiring high-field, narrowband THz pulses, such as pumping of Josephson plasmon resonances in layered superconductors[30–32]. Spectroscopic applications may be accessible with our source parameters[9], and we have demonstrated the technique at multiple frequencies. However, the lack of demonstrated tunability of the PPLN technology may limit the practical application to spectroscopic scenarios that do not require a large number of frequencies.

As chirp-and-delay is ubiquitously used for generating pulse trains, tuning the pulse-train is also relevant for a wide range of applications that require resonant excitation, including excitation of plasma waves[33], high-order harmonic generation[22], laser modulation of electron beams[23], coherent anti-Stokes Raman scattering microscopy of biological structures[24], excitation of atoms[34], molecules and solids and driving photocathode injectors for conventional accelerators[35]. These applications have so far been limited by the presence of high-order phase in the chirped-pulse drivers and have suffered from the lack of practical possibilities to remove it. Our technique addresses this need. In addition, the technique opens up the possibility of generating pulse trains with customised temporal periodicity profiles with novel applications such as generation of pre-chirped multicycle THz pulses that compress under the action of waveguide dispersion, and thus address a difficult problem in the delivery of high-field THz to an interaction point[36].

## Methods

**Laser system**. We use a 260-ps FWHM uncompressed, high-power CPA-based Ti: Sapphire laser providing 1.2 J at 5 Hz repetition rate. Inherent to the Öffner stretcher design, the laser pulses have a GDD of $\phi_2 = 2.05 \times 10^6$ fs$^2$ and a TOD of $\phi_3 = -4 \times 10^6$ fs$^3$ with additional uncharacterised fourth and higher order spectral phase. The beam has a super-Gaussian profile of 13 mm 1/e$^2$ intensity diameter, and a spectral width of more than 30 nm FWHM centred at 800 nm.

**Dispersion management**. Using the common description of spectral phase, the time-dependent frequency of a chirped pulse can be written as $\omega(t, \phi_2, \phi_3) = \omega_0 + 1/\phi_2 \times t - \phi_3/2\phi_2^3 \times t^2$, with $\omega_0$ the central frequency, and $\phi_2$ and $\phi_3$ the pulse GDD and TOD. Here, we neglect higher orders and assume the GDD is large. The parameters $\phi_2$ and $\phi_3$ are properties of the pulse and are typically inherent to the

laser design. When there is no TOD (i.e., $\phi_3 = 0$), the chirp is linear and the instantaneous difference in frequency $\Delta\omega$ is constant, determined only by the chirp rate (i.e., GDD, $\phi_2$) and the delay:

$$\Delta\omega(t) = \omega(t, \phi_2) - \omega(t - \Delta t, \phi_2) = \frac{\Delta t}{\phi_2}. \quad (1)$$

In the presence of TOD, however, the chirp becomes nonlinear (i.e., curved), and $\Delta\omega$ becomes a linear function of time with a slope that is proportional to the TOD:

$$\Delta\omega(t) = \omega(t, \phi_2, \phi_3) - \omega(t - \Delta t, \phi_2, \phi_3) = \left[\frac{\Delta t}{\phi_2} + \frac{\phi_3}{2\phi_2^3}\Delta t^2\right] - \left[\frac{\phi_3}{\phi_2^3}\Delta t\right] \times t. \quad (2)$$

Applying the condition $\Delta\omega = \Omega_{THz}$ shows that phase matching is only achieved at a particular time, $t^*$, which is dependent on the delay $\Delta t$ and given by:

$$t^*(\Delta t) = \frac{\phi_2^2}{\phi_3} + \frac{1}{2}\Delta t - \frac{\phi_2^3}{\phi_3\Delta t}\Omega_{THz}. \quad (3)$$

In the case of two identical chirped pulses, perfect phase matching requires elimination of the time-dependence of $\Delta\omega$, and thus can only be achieved by complete elimination of the TOD. Adjusting the relative dispersion of the two pulses, however, provides additional options. Adding a small amount of GDD, $\Delta\phi_2$, and TOD, $\Delta\phi_3$, reduces the curvature of the time-frequency mapping, which is equivalent to slightly tilting the pulse in time-frequency space, shown in Fig. 1c. The instantaneous difference in frequency between two chirped pulses, where one has been delayed by $\Delta t$ and tuned in spectral phase by adding $\Delta\phi_2$ and $\Delta\phi_3$, then reads

$$\Delta\omega(t) = \omega(t, \phi_2, \phi_3) - \omega(t - \Delta t, \phi_2 + \Delta\phi_2, \phi_3 + \Delta\phi_3). \quad (4)$$

It is straightforward to show that $\Delta\omega(t)$ is now a second order equation in $t$, expressible as $\Delta\omega(t) = c_0 + c_1 \times t + c_2 \times t^2$ with $c_i = c_i(\phi_2, \phi_3, \Delta\phi_2, \Delta\phi_3, \Delta t)$. More explicitly stated,

$$\Delta\omega(t) = \left[\frac{1}{\phi_2 + \Delta\phi_2}\Delta t + \frac{\phi_3 + \Delta\phi_3}{2(\phi_2 + \Delta\phi_2)}\Delta t^2\right]$$
$$+ \left[\frac{1}{\phi_2} - \frac{1}{\phi_2 + \Delta\phi_2} - \frac{\phi_3 + \Delta\phi_3}{(\phi_2 + \Delta\phi_2)^3}\Delta t\right] \times t \quad (5)$$
$$- \left[\frac{\phi_3}{2\phi_2^3} - \frac{\phi_3 + \Delta\phi_3}{2(\phi_2 + \Delta\phi_2)^3}\right] \times t^2.$$

According to this expression, a constant instantaneous frequency difference $\Delta\omega = \Omega_{THz}$, as illustrated in Fig. 1c, can be recovered by setting the coefficients $c_1$ and $c_2$ to zero, and solving for $\Delta\phi_2$, $\Delta\phi_3$, and $\Delta t$

$$\Delta\phi_2 = \phi_2\left(\frac{\phi_2}{\phi_2 - 2\phi_3\Omega_{THz}}\right)^{\frac{1}{2}} - \phi_2$$
$$\Delta\phi_3 = \phi_3\left(\frac{\phi_2}{\phi_2 - 2\phi_3\Omega_{THz}}\right)^{\frac{3}{2}} - \phi_3 \quad (6)$$
$$\Delta t = \frac{2\phi_2\Omega_{THz}}{1 + \left(1 - \frac{2\phi_3\Omega_{THz}}{\phi_2}\right)^{\frac{1}{2}}}.$$

**THz generation**. The chirp-and-delay concept is implemented using a Mach-Zehnder interferometer, with the GDD and TOD compensation distributed between the two interferometer arms. SF11 adds GDD of $\Delta\phi_2/L = +187.5$ fs$^2$/mm and additional small TOD of $\Delta\phi_3/L = +126$ fs$^3$/mm. Custom dielectric mirrors each add $\Delta\phi_3 = +20,000$ fs$^3$ and no additional GDD, independently confirmed with a white-light interferometer. For experiments at high fluence, two telescopes locally increase the beam size on the DCMs. The dispersion through the lenses is about 800 fs$^2$ and compensated with additional 4.4 mm SF11, which is not explicitly mentioned in the main text. The setup is very sensitive to the angular overlap of the two beams, which we ensure to sub-100 μrad level with a far field camera after the second beam splitter. The PPLN crystals are cooled below 100 K in a cryostat operating at $8 \times 10^{-4}$ mbar. The entrance window is AR coated for the NIR driver pulses. A ceramic aperture in front of the crystals (8.5 × 14.5 mm crystal #1, 9.5 × 14.5 mm crystal #2) prevents the beam from clipping on the crystal edges. We use Teflon plates with calibrated transmission, to separate the driver IR beam from the generated THz pulses. The THz energy is detected with a pyroelectric sensor (Gentec-EO: THZ9B-BL-BNC) that had been calibrated to a device calibrated by the Physikalisch-Technische Bundesanstalt (PTB). Each data point reported in the figures is averaged over 5 pulses (long delay scan, Fig. 4b), 100 pulses (GDD and TOD tuning, Fig. 3), and 20 pulses (fluence scan, Fig. 5). When error bars are not shown the error is below 5% rms and not visible within the given figure.

**PPLN crystals**. We use custom $10 \times 15$ mm$^2$ large aperture crystals[29] with a length of 36 mm and poling periods of 330 μm and 212 μm, corresponding to THz pulses of 361 GHz and 558 GHz, respectively. For the generation of high-energy THz pulses reported in Fig. 5, the 361 GHz crystal #1 is AR coated for the driver laser and features a 100-μm fused silica wafer at the end facet of the crystal to minimise Fresnel losses and maximise THz extraction. For comparison, the 361 GHz crystal

#2 is left uncoated and without a wafer installed. The AR coating and fused silica wafer together enhance the THz output of crystal #1 by about 35% compared to crystal #2. The remaining 25% difference in observed THz output is attributed to variations in the crystal manufacturing.

## Data availability
The data that support the findings of this study are available from the corresponding author upon reasonable request.

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

## Acknowledgements

We thank M. Schnepp for help with the laser system, M. Schust and T. Tilp for technical support on the experimental setup, P. Messner for assistance with the vacuum system, K. Ravi for theoretical discussions, and S.-H. Chia for help with the white-light inter-ferometry measurements. We acknowledge support from the European Research Council ERC Grant (609920), BMBF grant 05K16GU2, PIER project PIF-2017-67, and the DESY Strategy Fund. S.W.J. and V.L. acknowledge support by the European Regional Devel-opment Fund (ERDF) (CZ.02.1.01/0.0/0.0/15_008/0000162). H.I. and T.T. acknowledge support from the JST-Mirai Program Grant Number JPMJMI17A1, Japan.

## Author contributions

N.H.M. proposed the asymmetric compensation technique, with S.W.J. contributing to the development. S.W.J. developed the theoretical description of the concept. S.W.J. and F.A. conducted the experiments with assistance from V.L., T.E., and N.H.M.; S.W.J. and F.A. analysed the data with detailed input from N.H.M., A.R.M., and F.X.K.; A.L.C. facilitated the collaboration with H.I. and T.T., who provided the large aperture P.P.L.N. crystals. S.W.J., N.H.M., and A.R.M. wrote the manuscript with feedback from all authors. A.R.M. and F.X.K. led the project.

## Additional information

**Competing interests:** The authors declare no competing interests.

