## [Peer Review File · Nature Communications]

Reviewers' comments:

Reviewer #1 (Remarks to the Author):

The authors have performed the changes in the manuscript what I suggested and have answered my question appropriately.

I suggest publication.

Reviewer #2 (Remarks to the Author):

The manuscript is improved very much, and it can be understood that their main claim is realization of the high-energy (not high-field) narrowband THz generation. I could appreciate that the authors introduce the new technique which compensates for the high-order dispersion and realize the enhancement of conversion efficiency. However, I feel that the introduction part is not coherent with their main results, and I recommend that they revisit the introduction part and modify the manuscript before publication. My comments are as follows,

(1) In the introduction part, they describe that "Some emerging applications, however, such as waveguide-based electron acceleration [14], resonant pumping of materials [15] and narrowband spectroscopy [9], require high-field, narrow-bandwidth THz pulses for which the technology is much less developed". Certainly, experiments for "resonant pumping of materials [15]" requires both the high-field and narrow-bandwidth THz pulses. However, though they describe the bandwidth and pulse energy, there is no description about the estimation of field strength in the main text. They may say the point of their work is the realization of high-energy sources. However, for the experiment of "the resonant pumping of materials" the field strength is also very important. The excitation modes in the materials has finite dephasing time (mostly within tens of ps in solids) and thus not all of their long-pulse is used for the coherent excitation of modes. In this case, the field strength within the dephasing time is more important factor than the total energy to induce nonlinear responses.

(2) Also, their source with narrowband width seems useful for the experiment of "narrowband spectroscopy [9]". However, their source shows discrete frequency determined by the design of PPLN and is not enough tunable to take a spectrum. I don't know how they use their source for spectroscopy. And, is the reference of [9] appropriate as a reference here? The reference [9] is on the nonlinear spectroscopy, isn't it?

(3) I guess the most immediate application of their source is the "waveguide-based electron acceleration [14]". Although they cite the reference of [14] for it, the general readers do not know the importance of the "waveguide-based electron acceleration [14]", and thus cannot follow how much their source is useful for the application without any estimation. I recommend the authors to describe how much and what limitation their developed source breaks in the field of "waveguide-based electron acceleration [14]". It's good to discuss what kind of great results comes to realization in that research field by using their developed source at the discussion part of their manuscript.

4) They also say in the introduction that "In particular, relativistic acceleration of electrons calls for optically-synchronised...multi-millijoule pulses with MV/cm electric-field strengths, frequencies below 1 THz ... bandwidths below 1% [16,17], for which no technology currently exists". This sentence says the importance of both the field strength and narrow bandwidth for that application, and thus the estimation of field strength should be done somewhere else in the main text. If the present work is just a milestone for that actual application and there are no actual and concrete applications, I think the manuscript should be published in the specialized journal in optics even if their invention is very sophisticated.

Reviewer 1 comments

The authors have performed the changes in the manuscript what I suggested and have answered my question appropriately. I suggest publication.

We thank the reviewer for the suggestion of publication.

Reviewer 2 comments

The manuscript is improved very much, and it can be understood that their main claim is realization of the high-energy (not high-field) narrowband THz generation. I could appreciate that the authors introduce the new technique which compensates for the high-order dispersion and realize the enhancement of conversion efficiency. However, I feel that the introduction part is not coherent with their main results, and I recommend that they revisit the introduction part and modify the manuscript before publication. My comments are as follows,

We thank the reviewer for the mention of the improvement of the manuscript, and for the recommendations for improving the introduction even further before publication. As it is extremely important to properly introduce a work and place it in a greater context, we have addressed the reviewer's recommendations in a form that we saw appropriate.

We have mainly addressed the comments below by expanding and modifying the discussion section. We believe the introduction is meant to generally place the manuscript in context, and after presenting the full results the discussion section can place it more directly in context. We have studied the introduction in great detail, but we find that to better address the spirit of the reviewer's comments it was more appropriate to modify the specific portions of the manuscript being addressed, which happened to be rather in the results and discussion sections (pages 13 and 17). The comment from the reviewer that the introduction is "not coherent with their main results" is unfortunately too vague to specifically act on, where on the other hand we could directly address comments and concerns in other parts of the manuscript.

(1) In the introduction part, they describe that "Some emerging applications, however, such as waveguide-based electron acceleration [14], resonant pumping of materials [15] and narrowband spectroscopy [9], require high-field, narrow-bandwidth THz pulses for which the technology is much less developed". Certainly, experiments for "resonant pumping of materials [15]" requires both the high-field and narrow-bandwidth THz pulses. However, though they describe the bandwidth and pulse energy, there is no description about the estimation of field strength in the main text. They may say the point of their work is the realization of high-energy sources. However, for the experiment of "the resonant pumping of materials" the field strength is also very important.

We have added clarification of this, by providing an example field-strength calculation when the THz is tightly focused. The reviewer is correct that field strength is indeed an important quantity for various applications, but we believe such a calculation is better placed in the results section (page 13, line 216) rather than the introduction, directly after reporting the measured pulse energies. We have added the following text:

"If the pulses were to be combined, the 0.6 mJ and roughly 200 ps pulses could produce a field strength of ~ 18 MV/m when focused to a $1/e^2$ waist of 2 mm ($2.4 \times \lambda_{\text{THz}}$)."

Please see the response to point #3 for an additional field strength calculation especially relevant for the acceleration application, which we added to the discussion section.

The excitation modes in the materials has finite dephasing time (mostly within tens of ps in solids) and thus not all of their long-pulse is used for the coherent excitation of modes. In this case, the field strength within the dephasing time is more important factor than the total energy to induce nonlinear responses.

Although this may indeed be the case with certain materials, we believe such a nuanced discussion may be outside the realm of this work. Please note that we do not claim our source might be directly applicable as is for the specific application of resonant pumping. Our intention is rather to provide a wider context for the relevance and need of a high-field narrowband THz source.

In fact, when discussing with colleagues the possibility of applying our source for resonant pumping, we concluded that with such a strong narrowband driver it may be that dynamics are different than in previous experiments.

In conclusion, we agree with the reviewer, that for resonant pumping in solids the dephasing time is a very relevant quantity. But as our intention is only to highlight the manifold applications that could in principle benefit from a high-field narrowband source, we decided to not further expand the discussion on resonant pumping in our manuscript.

In addition, we believe the inclusion of the field strength as described above is a good opportunity to provide greater quantitative prospects for experiments in solids.

(2) Also, their source with narrowband width seems useful for the experiment of “narrowband spectroscopy [9]”. However, their source shows discrete frequency determined by the design of PPLN and is not enough tunable to take a spectrum. I don’t know how they use their source for spectroscopy. And, is the reference of [9] appropriate as a reference here? The reference [9] is on the nonlinear spectroscopy, isn’t it?

Indeed the reviewer is correct that although our parameters are generally useful for spectroscopic applications, the lack of demonstrated tunability with a single crystal limits the practical application for certain specific forms of spectroscopy (for example absorption spectroscopy with many narrow frequencies). Although tunability may be possible with a single PPLN crystal – and this is a subject of ongoing investigation – we do not discuss it in the manuscript. In order to address this, we add the following text to the discussion section (page 17, line 298) to weaken/limit our claim:

“Spectroscopic applications may be accessible with our source parameters [9], and we have demonstrated the technique at multiple frequencies. However, the lack of demonstrated tunability of the PPLN technology may limit the practical application to spectroscopic scenarios that do not require a large number of frequencies.”

We must note that in our use of the term spectroscopy we are operating on the more generic definition of the term corresponding to basic light-matter interaction, which includes studying the spectral response of a sample (absorption, transmission, etc, etc), but does not necessarily require a large number of pump frequencies. We hope that the specification above, that the practical application of our source may be limited to spectroscopic applications that do not require a large number of frequencies, is helpful in clarifying this.

Lastly, indeed we do intend to reference nonlinear spectroscopy, since the parameters demonstrated would be sufficient to induce nonlinear responses. This is surely dependent on the intended sample to be studied, but our main purpose of mentioning this application is to provide a context for our high energy THz source, and the direct applicability to *any* specific spectroscopic scenario is beyond the scope of the discussion.

(3) I guess the most immediate application of their source is the “waveguide-based electron acceleration [14]”. Although they cite the reference of [14] for it, the general readers do not know the importance of the “waveguide-based electron acceleration [14]”, and thus cannot follow how much their source is useful for the application without any estimation. I recommend the authors to describe how much and what limitation their developed source breaks in the field of “waveguide-based electron acceleration [14]”. It’s good to discuss what kind of great results comes to realization in that research field by using their developed source at the discussion part of their manuscript.

We elaborate on this point, by adding the expected field strength and acceleration in a waveguide accelerating structure, and comparing it to past results where the THz driving the acceleration was less than ideal. We added the following text (to the discussion section, page 17, line 286) to that effect:

“It is expected that 0.6 mJ of THz in a 200 ps long pulse as produced in this work could create a field strength of 50 MV/m when coupled in to a waveguide designed for matched acceleration. With acceleration over the roughly 4 cm allowed by such a pulse, sub-relativistic electrons could be accelerated beyond 2 MeV, an energy already relevant for ultrafast electron diffraction. This can be compared to past acceleration in a waveguide with broadband THz, where the acceleration length was limited to 3 mm and the energy gain was limited to 7 keV [1].”

4) They also say in the introduction that “In particular, relativistic acceleration of electrons calls for optically-synchronised...multi-millijoule pulses with MV/cm electric-field strengths, frequencies below 1 THz ... bandwidths below 1% [16,17], for which no technology currently exists”. This sentence says the importance of both the field strength and narrow bandwidth for that application, and thus the estimation of field strength should be done somewhere else in the main text. If the present work is just a milestone for that actual application and there are no actual and concrete applications, I think the manuscript should be published in the specialized journal in optics even if their invention is very sophisticated.

This is not entirely a separate comment from points #1 and #3, addressed above with two separate calculations of the field strength. Here, we can only stress again that there are many concrete applications of such high energy THz, and beyond that the application of the spectral phase tuning concept to other optical processes. These applications beyond THz generation are referenced significantly in the introduction and discussion sections.

The applications that we chiefly reference are 1) electron acceleration, for which we have now provided concrete numbers communicating the direct relevance, 2) nonlinear spectroscopy, for which we have clarified in response to the reviewer’s comment, and 3) resonant pumping of materials.

The reviewer’s comments on application 2 and 3 are surely accurate, in the sense that our high energy narrowband THz source may not be directly applicable to *some* scenarios. However, we strongly believe that the cutting-edge nature of the source means that although it may not

apply to *every* application, it will be applicable to many scenarios beyond our expertise due to the new regime it would explore.

Beyond the applications of the THz source we have demonstrated, the general nonlinear optical technique of spectral phase tuning has never been shown before, and this may find applications in atomic, molecular, or optical physics beyond THz generation. This may include generating plasma waves, controlling molecules, certain forms of NIR spectroscopy, and beyond. We believe that the combination of a new nonlinear optical technique and the demonstrated application of mJ-scale THz generation (which is cutting-edge by itself) are the crowning achievements of this work.

REVIEWERS' COMMENTS:

Reviewer #2 (Remarks to the Author):

My concerns have been perfectly addressed. I recommend the revised manuscript should be accepted for publication in Nature Communications.